# Accurate detection and quantification of SARS-CoV-2 genomic and subgenomic mRNAs by ddPCR and meta-transcriptomics analysis

Annarita Oranger [1,7], Caterina Manzari[1,7], Matteo Chiara[2,3,7], Elisabetta Notario[1], Bruno Fosso [2], Antonio Parisi[4], Angelica Bianco[4], Michela Iacobellis[5], Morena d'Avenia[5], Anna Maria D'Erchia [1,2,6✉] & Graziano Pesole [1,2,6✉]

SARS-CoV-2 replication requires the synthesis of a set of structural proteins expressed through discontinuous transcription of ten subgenomic mRNAs (sgmRNAs). Here, we have fine-tuned droplet digital PCR (ddPCR) assays to accurately detect and quantify SARS-CoV-2 genomic ORF1ab and sgmRNAs for the nucleocapsid (N) and spike (S) proteins. We analyzed 166 RNA samples from anonymized SARS-CoV-2 positive subjects and we observed a recurrent and characteristic pattern of sgmRNAs expression in relation to the total viral RNA content. Additionally, expression profiles of sgmRNAs, as determined by meta-transcriptomics sequencing of a subset of 110 RNA samples, were highly correlated with those obtained by ddPCR. By providing a comprehensive and dynamic snapshot of the levels of SARS-CoV-2 sgmRNAs in infected individuals, our results may contribute a better understanding of the dynamics of transcription and expression of the genome of SARS-CoV-2 and facilitate the development of more accurate molecular diagnostic tools for the stratification of COVID-19 patients.

[1] Department of Biosciences, Biotechnology and Biopharmaceutics, University of Bari Aldo Moro, Via Orabona 4, 70126 Bari, Italy. [2] Institute of Biomembranes, Bioenergetics and Molecular Biotechnologies, National Research Council, Via Amendola 122/O, 70126 Bari, Italy. [3] Department of Biosciences, University of Milan, Via Celoria 26, 20133 Milan, Italy. [4] Istituto Zooprofilattico Sperimentale della Puglia e Basilicata, 70017 Putignano (Bari), Italy. [5] Servizio Centralizzato Aziendale di Citopatologia e Screening- PO "Di Venere" - ASL, 70131 Bari, Italy. [6] Consorzio Interuniversitario Biotecnologie, 34100 Trieste, Italy. [7] These authors contributed equally: Annarita Oranger, Caterina Manzari, Matteo Chiara. ✉email: annamaria.derchia@uniba.it; graziano.pesole@uniba.it

The COVID-19 (Coronavirus Disease 2019) outbreak caused by severe acute respiratory syndrome-coronavirus 2 (SARS-CoV-2) hit the world with a global pandemic. More than 197 million of confirmed infections and more than 4,200,000 deaths have been recorded worldwide since the first reported case in late December 2019, (WHO Coronavirus Disease (COVID-19) Dashboard; data last updated: 2021/29/7).

SARS-CoV-2 is an enveloped positive-sense single-stranded RNA betacoronavirus. The genome sequence (gRNA) is ~30 kb in size and shows the typical arrangement of betacoronavirus genomes. The replicase gene, which consists of two long, overlapping open reading frames encoding for polyproteins (ORF1a and ORF1b) extends over the 5′ proximal two-thirds of the genome, while the 3′ terminal region of the genome encodes 4 structural proteins required for the assembly of the viral capsid: spike (S), envelope (E), membrane (M) and nucleocapsid (N) and 8 other less well-characterized proteins, which are not universally conserved among coronaviruses.

SARS-CoV-2 infection initiates with the attachment of the virion to the surface of target cells, mediated by the binding of the S glycoprotein to the angiotensin-converting enzyme 2 (ACE2) receptor[1]. Proteolytic cleavage of the S protein by a cathepsin (TMPRRS2 or other proteases), followed by the fusion of the viral and cellular membranes, allows the entry of the virus in the cytosol[2,3]. Subsequently, the gRNA is translated to produce the polyproteins that are post-translationally processed by viral encoded proteases to produce 16 non-structural proteins (nsp) whose function is related to the synthesis of viral genomic RNA and to escape immune response[4–6].

Structural and accessory proteins are translated from a set of nested transcripts, called subgenomic mRNAs (sgmRNAs). These sgmRNAs are produced during viral genome replication through a complex template-switching discontinuous transcription mechanism, which occurs during the synthesis of the negative-stranded RNA and is mediated by short, conserved transcriptional regulatory sequences (TRSs) that punctuate the viral genome and are found upstream of each major ORF (TRS-B) and in the 5′ UTR (TRS-L)[7]. Briefly, when the replicase complex encounters a TRS-B element during the elongation of nascent minus-strand RNA, the complementarity with the TRS-L in the 5′ UTR can mediate a relocation of the complex to the 5′ UTR, resulting in the synthesis of a discontinuous negative sense sgmRNA. Transcription of negative sense sgmRNAs generates positive sense sgmRNAs, encoding for S, E, M, N, and accessory proteins, which are 5′ and 3′ coterminal with the gRNA[8–10].

The reverse transcription-quantitative polymerase chain reaction (RT-qPCR) is considered the current gold standard method for the diagnosis of COVID-19. Although standard tests for COVID-19 are based on naso-pharyngeal swab samples, in principle RT-qPCR allows the detection of SARS-CoV-2 genomic RNA from different types of specimens[11–15]. This method provides a relative quantification of the total amount of viral RNAs, expressed in the form of a threshold cycle (Ct) value, which represents the PCR cycle number at which the target product crosses the threshold for detection. Although different probes/primer sets have been reported to have different levels of sensitivity, a Ct cut-off value of positivity (usually 40) has been established for all approved commercial SARS-CoV-2 molecular diagnostic kits[16]. The major limitation of commonly-used SARS-CoV-2 diagnostic RT-qPCR approaches is that, with the exception of the ORF1ab, which does not require a specific mechanism of transcription and is equimolar with the genome, they detect all viral RNA unspecifically, regardless of the nature and function (genomic, subgenomic or even degradation products). Indeed, this approach can not discriminate between sgmRNA molecules and the genomic sequence of the corresponding gene. The net result is that the Ct value derives from the sum of genomic and subgenomic RNAs. On the other hand, since sgmRNAs are transcribed in infected cells and encode for structural viral proteins, assembling in virion particles, their specific detection could provide evidence of active viral replication rather than the possible presence of residual viral RNA.

In this work, we evaluated the application of droplet digital (ddPCR) assay to perform a proof of concept study for the detection and quantification of SARS-CoV-2 sgmRNAs coding for S and N proteins from RNA samples obtained from nasopharyngeal swabs of SARS-CoV-2 positive subjects. By analyzing 166 RNA samples, we derived some relevant observations concerning the patterns of expression of SARS-CoV-2 sgmRNAs in infected subjects, with possible implications also in diagnostics. Firstly, we found that, while sgmRNA copy numbers are positively correlated with the total number of gRNA copies, N and S sgmRNAs have a characteristic range of expression, which remains similar across all samples, regardless of the viral RNA content. Additionally, we observed that sgmRNAs expression levels are reduced in RNA samples with a low viral RNA content, thus indicating that these samples are mainly characterized by residual genomic SARS-CoV-2 material with scarce or no active viral transcription.

Moreover, by performing bioinformatics analyses of meta-transcriptomics sequencing of a subset of 110 RNA samples, we demonstrated that sgmRNA expression patterns recovered by NGS sequencing are highly consistent with those inferred by ddPCR, thus suggesting that this approach can provide an accurate overview of the expression patterns of SARS-CoV-2 sgmRNAs and extend our understanding of the mechanism of transcription of the genome of this novel pathogen.

Finally, since we observed a good correlation between N and S sgmRNAs expression levels, as derived from sequencing data, and their copy number, as determined by ddPCR (although less significant for the S sgmRNA), we envisage that this and other similar/equivalent approaches could also be used to derive a realistic estimate of the viral load associated with SARS-CoV-2 RNA samples for which meta-transcriptomic data are available.

## Results

**Samples used in this study.** For this study, RNA samples extracted from 166 nasopharyngeal swab remnants of unidentified subjects with a positive diagnosis of COVID-19 and 14 nasopharyngeal swab remnants of unidentified subjects with a negative diagnosis of COVID-19 were used. No clinical data were available, but only age and gender (Table 1 and Supplementary Data 1). The presence of viral RNA in positive samples was confirmed by a RT-qPCR assay (see "Methods" section).

**Assessment of specificity and reproducibility of designed SARS-CoV-2 ddPCR assay for genomic and subgenomic mRNAs.** Two pairs of primers were designed to quantify S and N sgmRNAs by ddPCR, with a forward primer targeting the LS sequence in the 5′UTR and a reverse primer in the 5′-proximal end of the S and N gene, respectively. A pair of primers in the ORF1ab (nsp8 sequence) was also designed to quantify gRNA levels. Primer pairs sequences and their genomic coordinates are reported in Table 2.

To evaluate the specificity of the three primers pairs, ddPCR assays were performed for ORF1ab RNA and S and N sgmRNAs using no template controls (NTC samples) and SARS-CoV-2 negative RNA samples (Supplementary Fig. 1). The false-positive rates (FPR) calculated on 20 NTC samples was 0.0067% for ORF1ab (1 positive droplet per sample detected in 2 out of 20 NTC samples), 0.0057% for N sgmRNA (1 positive droplet per

**Table 1 Summary information about SARS-COV-2 positive RNA samples and individuals included in this study.**

|  | High viral RNA content* | Middle viral RNA content* | Low viral RNA content* |
|---|---|---|---|
| SARS-COV-2 positive RNAs n. | 21 | 82 | 63 |
| Patient's gender | 9 F, 12 M | 34 F, 48 M | 32 F, 31 M |
| Patient's age (years mean ± SD) | 49 ± 20 | 55 ± 22 | 44 ± 20 |
| NGS sequenced RNAs | 21 | 60 | 31 |

*RNA samples were grouped applying a hierarchical agglomerative clustering algorithm to ddPCR absolute quantification of SARS-CoV-2 genomic and sub-genomic RNAs.

**Table 2 Primer pairs designed in this study to target specific SARS-CoV-2 RNAs.**

| SARS-CoV-2 target | Primer name | Sequence (5′→3′) | Amplicon length | Genome coordinates (NC_045512v2) |
|---|---|---|---|---|
| ORF1ab RNA | ORF1ab For | AGATCTGAGGACAAGAGGGCA | 158 bp | 12314−12334 |
|  | ORF1ab Rev | TTGGCTGCTGTTGTAAGAGGTA |  | 12471−12450 |
| N sgmRNA | LS_For1 | CGATCTCTTGTAGATCTGTTC | 205 bp | 44−64 |
|  | N_Rev | AGCGGTGAACCAAGACGCA |  | 28438−28420 |
| S sgmRNA | LS_For2 | CCAACTTTCGATCTCTTGTAG | 207 bp | 36−56 |
|  | S_Rev | AGAACAAGTCCTGAGTTGAATG |  | 21728−21707 |

sample detected in 3 out of 20 NTC samples) and 0.0063% for S sgmRNA (1 positive droplet per sample detected in 2 out of 20 NTC samples). FPR calculated on 14 SARS-CoV-2 negative RNA samples was 0.0087% for ORF1ab (1 positive droplet per sample detected in 4 out of 14 SARS-CoV-2 negative samples), 0.0073% for N sgmRNA (1–2 positive droplets per sample detected in 11 out of 14 SARS-CoV-2 negative samples) and 0.0103% for S sgmRNA (1–2 positive droplets detected in 2 out of 14 SARS-CoV-2 negative samples). All data are reported in Supplementary Data 2.

To assess whether our assays could specifically detect sgmRNAs and not the corresponding genomic sequences, ddPCR assays were also performed using as templates custom synthesized DNA sequences corresponding to a portion of each SARS-CoV-2 target transcript (ORF1ab, N and S sgmRNA) and DNA sequences corresponding to the 5′ region of N and S genes (Supplementary Table 1). Amplification was detected only when DNA sequences corresponding to SARS-CoV-2 sgmRNAs were used (Supplementary Fig. 1).

The sequences of the ORF1ab, N, and S sgmRNAs amplicons obtained from the amplification on SARS-CoV-2 positive RNA samples (RNA122 for N sgmRNA and RNA 25 for S sgmRNA) were also confirmed by Sanger sequencing (Supplementary Fig. 2).

The reproducibility of our ddPCR assays was evaluated by calculating intra- and inter-assay coefficient of variation (CV%) using a DNA template corresponding to a portion of ORF1ab gRNA and N and S sgmRNAs. Intra-assay CV (%) was 3.2, 4.4, and 7.4% respectively for ORF1ab, N, and S sgmRNAs; inter-assay CV (%) was 9.7, 8.5, and 11.98% respectively for ORF1ab, N, and S sgmRNAs (Supplementary Data 3).

These results indicate that our ddPCR assays are highly specific and reproducible in detecting ORF1ab RNA and N and S sgmRNAs.

**Accuracy and limit of detection (LoD) of ddPCR assays targeting genomic and subgenomic mRNAs SARS-CoV-2.** To evaluate the accuracy of our ddPCR assays, we used the DNA templates corresponding to a portion of each SARS-CoV-2 target transcript. Ten serial dilutions of each of these templates were used in ddPCR (Supplementary Data 4–6). As shown in Fig. 1a–c, we found a very good linear correlation between the expected and observed copy numbers for the three SARS-CoV-2 targets ($R^2 = 1$, 0.998, and 0.992 respectively for ORF1ab and N and S

sgmRNAs). The Limit of Detection (LoD) of each ddPCR assay was assessed also by analyzing the same serial dilutions used for the evaluation of the accuracy (Supplementary Data 4–6). Likewise, LoD, which is defined as the limit of an analyte/target detectable by a molecular assay with 95% of confidence and generally expressed as copies/reaction, was calculated for each target by regression probit analysis. As shown in Fig. 1d–f, LoD was 1.18 (95% CI: 0.66–2.08), 2.3 (95% CI: 1.05–11.8) and 5.2 (95% CI: 2.8–20.1) copies/reaction, respectively for ORF1ab, N and S sgmRNAs.

These results demonstrate that our ddPCR assays are accurate and have high sensitivity in detecting ORF1ab RNA and N and S sgmRNAs.

**Absolute quantification of SARS-CoV-2 ORF1ab mRNA and N and S sgmRNAs by ddPCR.** ddPCR assays for the quantification of SARS-CoV-2 ORF1ab gRNA and of the N and S sgmRNAs were applied to 166 RNA samples obtained from naso-pharyngeal swabs of SARS-CoV-2 positive subjects (Supplementary Data 7). Overall, a high variability in copy numbers was observed among the RNA samples, for all the three targets. ORF1ab was consistently associated with a higher number of RNA copies, compared to N and S sgmRNAs with S sgmRNA consistently associated with the lowest copy number in all samples. Based on observed gRNA and sgmRNAs copy numbers (see "Methods"), samples were stratified into three main groups: "high" ($n = 21$), "middle" ($n = 82$) and "low" ($n = 63$) viral RNA content. Interestingly, both N and S sgmRNAs displayed a higher number of copies in samples of the "high" group (N sgmRNA median = 49,060 copies/ng RNA, 25° percentile = 20,463, 75° percentile = 199,650; IQR = 179,187; S sgmRNA median = 3,953 copies/ng RNA, 25° percentile = 1,173, 75° percentile = 13,266, IQR = 12,093) and "middle" group (N sgmRNA median = 387 copies/ng RNA, 25° percentile =119.3, 75° percentile = 1,386, IQR = 1,266; S sgmRNA median = 27.5 copies/ng RNA, 25° percentile = 7.725, 75° percentile = 88, IQR = 80.2), while in samples belonging to the "low" group, both N and S subgenomic transcripts were barely detectable (N sgmRNA median = 6.2 copies/ng RNA, 25° percentile = 2.8, 75° percentile = 27, IQR = 24.2; S sgmRNA median = 0 copies/ng RNA, 25° percentile = 0, 75° percentile = 1.8, IQR = 1.8). In particular, N sgmRNA was absent or scarcely detectable (<1 copy/ng RNA) in 12 out of 15 samples with less than 10 copies/ng RNA of ORF1ab, while S sgmRNA

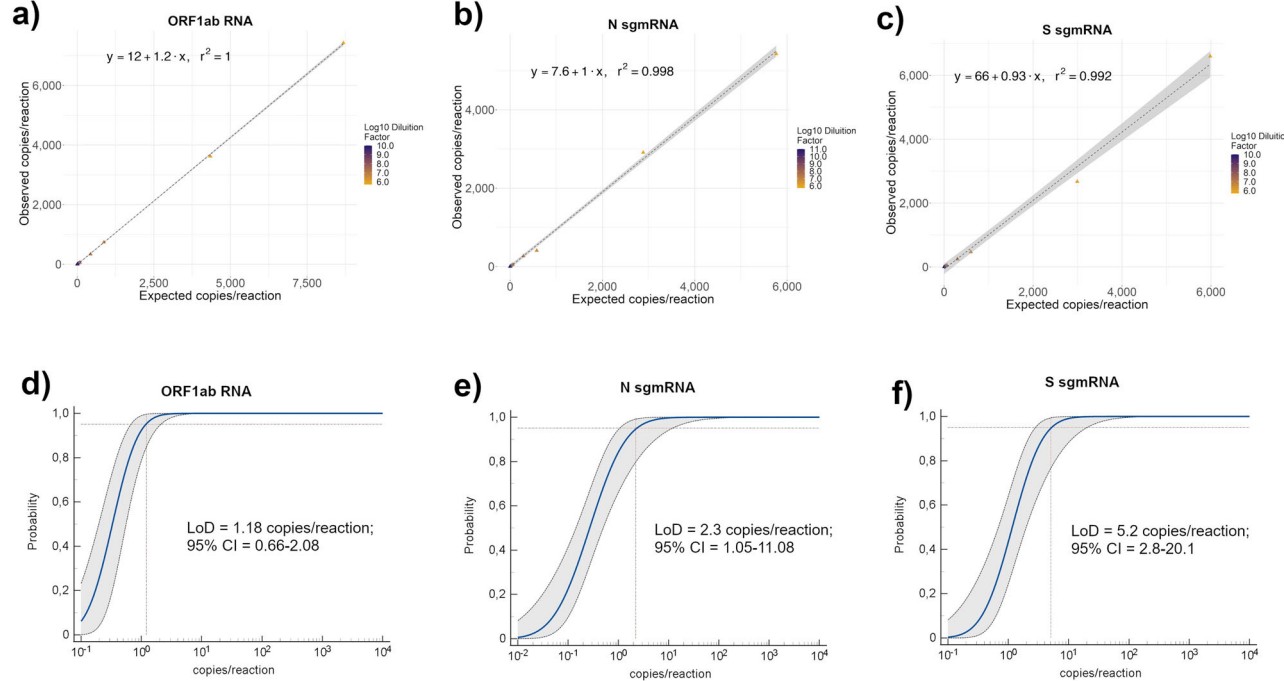

**Fig. 1 Accuracy and Limit of detection (LoD) of SARS-CoV-2 genomic ORF1ab RNA and N and S sub-genomic RNAs by ddPCR assays. a–c** Accuracy of designed SARS-CoV-2 ddPCR assays was evaluated using ten serial dilutions, each with at least nine replicates, performed on custom synthetic DNA sequences corresponding to portions of ORF1ab RNA and N and S sgmRNAs. Accuracy was evaluated by linear regression analysis plotting expected (Y-axis) *vs* ddPCR observed (X-axis) copies per reaction for ORF1ab RNA (**a**), N sgmRNA (**b**) and S sgmRNA (**c**). The regression equation and $R^2$ are shown in the top corner. The gray area represents a 0.95 level of confidence interval. **d–f** Limit of detection (LoD) of designed SARS-CoV-2 ddPCR assays was evaluated on the same serial dilutions of custom synthetic DNA sequences corresponding to portions of ORF1ab RNA and N and S sgmRNAs, used for the accuracy analysis. LoD for ORF1ab (**d**), N sgmRNA (**e**), and S sgmRNA (**f**) was defined by probit analysis using MedCalc software.

was absent or scarcely detectable in 36 out of 45 samples with less than 100 copies/ng RNA of ORF1ab.

As shown in Fig. 2a, b, N and S sgmRNAs progressively decreased in parallel with total viral RNA content. All sgmRNAs displayed the same expression trend with statistically significant copy number differences between the three groups of samples (*p*-value < 0.0001).

Ratios of N and S sgmRNAs to viral genome copy numbers (ORF1ab RNA target) were calculated for the three groups of samples with different viral RNA content. As shown in Fig. 2c, d, N and S sgmRNAs had a significant higher expression ratio in the "high" group compared with both the "middle" and "low" group (N: *p*-value = 0.01075 and 0.001314 for low and middle respectively; S: *p*-value = 0.002126 for low and 0.0001246 for middle) suggesting increased levels of sgmRNAs (and potentially of increased viral transcription) in subjects with high SARS-CoV-2 gRNA content. Importantly, no significant differences were observed when the low and middle groups were compared. As reported in Fig. 2c, d, samples in the "low" group were characterized by higher levels of variability (see the extent of the whiskers in the box plot) compared with the other groups; probably due to the fact being based on a reduced number of copies (45.8, 16.2 and 1.2 on average for ORF1ab, N and S respectively), estimates obtained for this group have a higher relative variability.

Based on the assumption that ~20 picograms of RNA are the average content of a mammalian cell[17], the expected number of N and S sgmRNAs per cell were calculated. These estimates of sgmRNAs copy number per cell can provide useful information about the intracellular viral content. As shown in Table 3, when the sgmRNAs average copies number/ng of input RNA were considered, samples belonging to the "high" group may contain

an average of 1,727 and 174 molecules of respectively N and S sgmRNAs per cell. The number of sgmRNAs copies per cell decreased in the "middle" group (~25 and 2 molecules per cell respectively for N and S sgmRNAs), while in the "low" group, on average, a cell may contain any copy of N and S sgmRNAs.

Overall, these data demonstrate that absolute copy number of SARS-CoV-2 gRNA and sgmRNAs are correlated and that, in specimens with very low viral load, as inferred from the gRNA copy number, sgmRNAs are scarcely or not present, thus providing indications of a possible absence of active viral replication in these samples.

**Meta-transcriptomics analysis shows a high correlation with ddPCR estimates of sgmRNAs expression.** A subset of the 166 RNA samples used in this study (113 samples) was subjected to meta-transcriptomics sequencing. Since meta-transcriptomic sequencing provides an accurate representation of all the RNA species present in a sample, these data were also used for deriving an estimate of the expression levels of SARS-CoV-2 sgmRNAs. A method based on the count of sgmRNA junctions spanning reads was applied (see "Methods" section). gRNA abundance was estimated by counting only reads covering the same genomic region in ORF1ab (nsp8 gene) used to estimate gDNA copy number in ddPCR assay. Only samples with 100 or more viral reads (110 out of 113 sequenced samples, Supplementary Data 8) were considered for the study of transcriptional profiles. As shown in Fig. 3, highly significant levels of correlation were observed when expression levels, as estimated by analysis of NGS meta-transcriptomic data, were compared with copy number estimates obtained by ddPCR, for all the three target genes (ORF1ab, Fig. 3a; N, Fig. 3b; S, Fig. 3c) included in our assay.

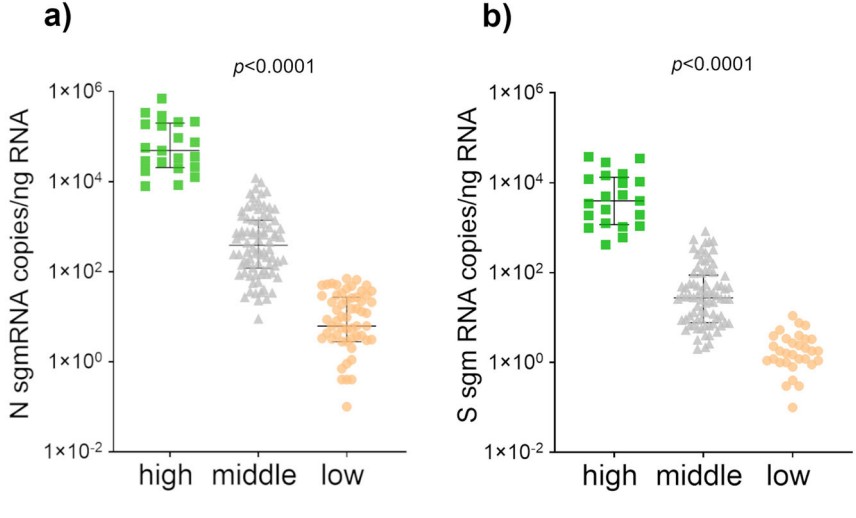

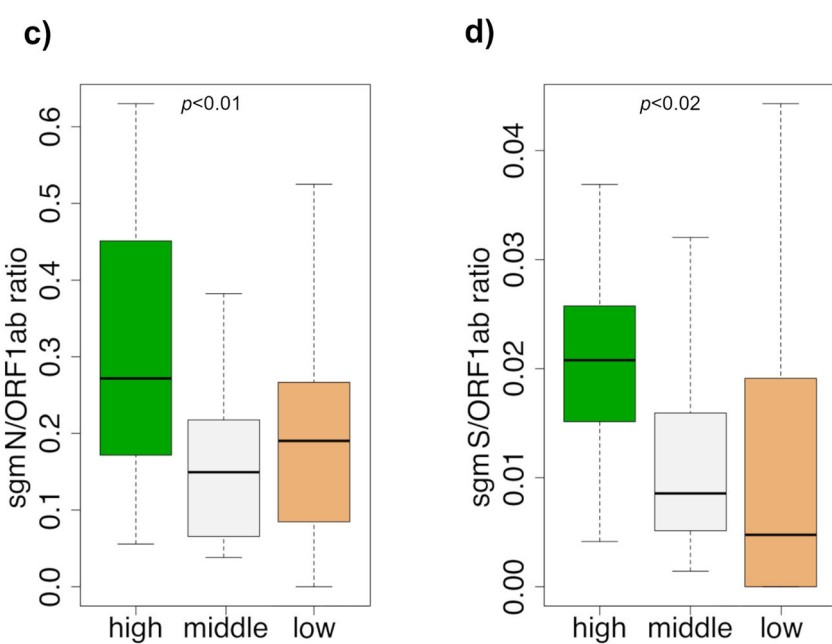

**Fig. 2 SARS-CoV-2 N and S sub-genomic RNAs absolute quantification and their relative expression to genomic RNA in SARS-CoV-2 positive samples by ddPCR.** N sgmRNA (**a**) and S sgmRNA (**b**) copies quantification in 166 RNA samples from SARS-CoV2 positive subjects. Based on ORF1ab gRNA and N and S sgmRNAs quantification, RNA samples were divided into three groups defined as "high" ($n = 21$, represented by the green square symbol), "middle" ($n = 82$, represented by the gray triangle symbol) and "low" ($n = 63$, represented by the orange circle symbol) viral RNA content. Values, reported in copies/ng RNA, are expressed as the means of a duplicate assay for each sample. Samples with zero sgmRNAs copies/ng RNA were not plotted in the graphs because logarithmic axes mathematically do not contemplate 0 value. Black lines represent median with relative interquartile range (IQR); $p$-value < 0.0001 calculated by Kruskal−Wallis test. N sgmRNA (**c**) and S sgmRNA (**d**) expression were evaluated as ratio respect to ORF1ab gRNA in "high" (green), "middle" (gray), and "low" (orange) viral RNA content group. Values are reported as sgmRNA/ORF1ab RNA copies/ng RNA. Significance was evaluated comparing "high" group *vs* "middle" and "low" groups by Mann−Whitney U test /Wilcoxon rank-sum. Data are presented as median with relative interquartile range (IQR). For N sgmRNA: $p$-value = 0.01075 for "low" and 0.001314 for "middle"; for S sgmRNA: $p$-value = 0.002126 for "low" and 0.0001246 for "middle".

Lower, but still very significant, levels of correlation were observed for the S sgmRNA with respect to other targets ($R^2 = 0.79$ for ORF1ab mRNA; $R^2 = 0.78$ for N sgmRNA; $R^2 = 0.66$ for S sgmRNA; $p < 0.0001$ for all targets). Consistent with ddPCR data, we observed that the viral gRNA was consistently associated with an increased (more than five-fold on average) number of reads if compared to sgmRNAs (Fig. 4).

When expression patterns of all sgmRNAs were compared, we also observed that, as previously noted in Kim et al.[18], sgmRNAs encoded by the most 3′ end terminal portion of the genome, displayed higher levels of expression, with the N sgmRNA showing the highest number of reads in all samples herein considered. Conversely, sgmRNAs for the S and ORF3a proteins show ~3 fold decrease in the number of reads if compared with

| Viral RNA content group | genomic ORF1ab (copies/ng RNA*) | N sgmRNA (copies/ng RNA*) | S sgmRNA (copies/ng RNA*) | N sgmRNA copies/cell | S sgmRNA copies/cell |
|---|---|---|---|---|---|
| High | 467,936 | 86,334 | 8,702 | 1,727 | 174 |
| Middle | 8,864 | 1,251 | 87 | 25 | 1.7 |
| Low | 74 | 16 | 1 | 0.3 | 0.02 |

*Values are reported as the mean of copies/ng RNA values.

**Table 3 N and S sgmRNA molecules number per cell according to the viral RNA content group.**

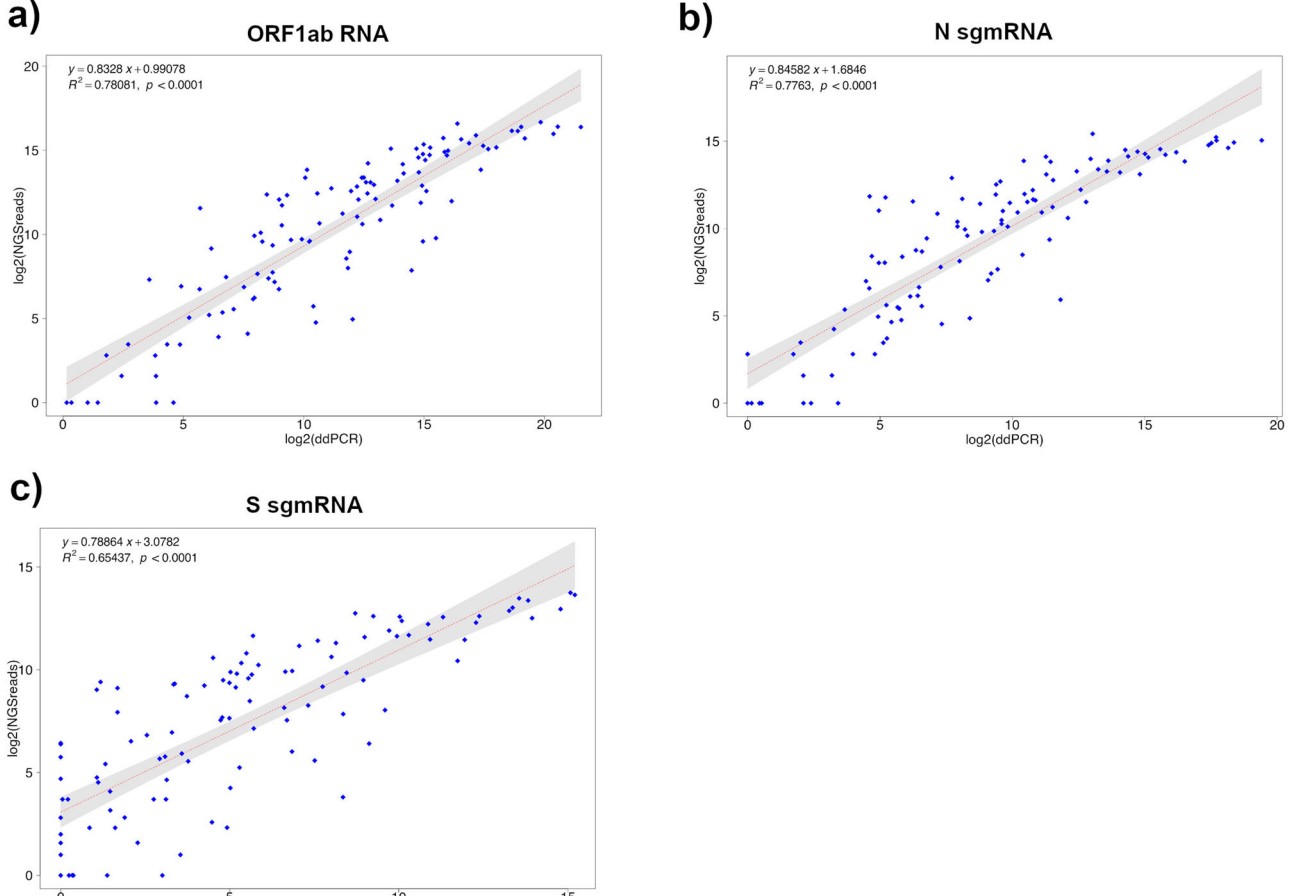

**Fig. 3 Correlation between ddPCR and transcriptomics data for SARS-CoV-2 ORF1ab, N and S sgmRNA targets.** Linear correlation between ddPCR and sequencing quantification of SARS-CoV-2 ORF1ab gRNA (**a**), N sgmRNA (**b**), and S sgmRNA (**c**) calculated on 110 SARS-CoV2 positive RNA samples, by means of a bivariate linear fit analysis (*p*-value < 0.0001). On *X*-axis: log2 scaled ddPCR quantification; on *Y*-axis: log2 scaled meta-transcriptomics reads count. The regression equation and $R^2$ are reported in the top-left corner. The gray area represents a 0.95 level of confidence interval.

the average of other sgmRNAs. Importantly all the sgmRNAs displayed characteristic abundance/level of expression which was largely consistent across all samples (Supplementary Fig. 3). Furthermore, by NGS analysis and consistently with our ddPCR data of N and S sgmRNAs relative expression, a significantly higher subgenomic to genomic RNA ratio was observed for all canonical sgmRNAs, in the "high" RNA content group if compared to "middle" and "low" groups (Supplementary Fig. 4). This is consistent with higher levels of viral transcription and replication in samples with high viral load.

Importantly, the remarkable levels of agreement observed both with ddPCR data and with previous studies for the characterization of expression patterns of the SARS-CoV-2 genomes[18] suggested that the approach developed for the quantification of sgmRNAs in this study is accurate, indicating that estimates of gRNA and sgmRNAs copy numbers from meta-transcriptomics

sequencing of SARS-CoV-2 positive samples, could be used as a proxy to provide a reasonable estimate of both genomic and subgenomic viral RNA content.

## Discussion

Qualitative and quantitative measures of viral load in SARS-CoV-2 positive patients provide effective tools for monitoring the progression of the COVID-19 and the severity of the infection. Changes in viral load are observed during the progression of the infection, reaching the peak during the first 4–6 days from the onset of the infection with a following gradual decrease[19]. Wölfel et al. reported that 8 days after the onset of the symptoms, SARS-CoV-2 isolated from positive patients failed to infect cell cultures, demonstrating a decline in virus infectiousness while Bullard and colleagues showed that infectivity (defined by virus growth in VERO cell culture) is remarkable only when RT-qPCR Ct value is

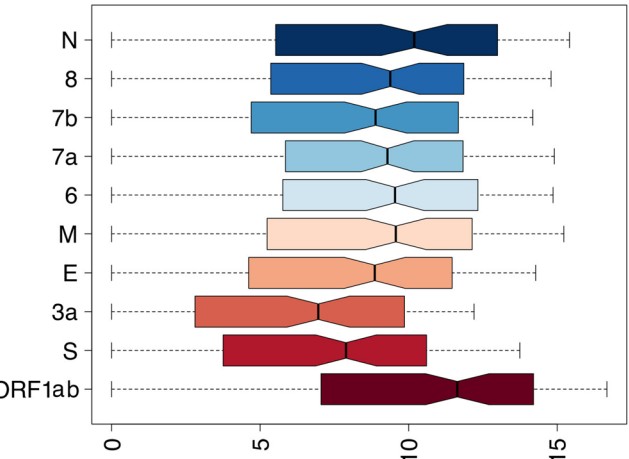

**Fig. 4 SARS-CoV-2 transcripts expression derived from meta-transcriptomics sequencing.** Representation by boxplots of log2 scaled counts distributions of meta-transcriptomics reads assigned to each sgmRNAs and to the ORF1ab gene (gRNA). Genes are indicated on the Y-axis. Log2 scaled counts on the X-axis.

lower than 24[20,21]. While recent studies reported detectable levels of viral shedding both in asymptomatic and symptomatic SARS-CoV-2 positive patients even after weeks from symptom resolution[22], the correlation between detectable viral RNA and transmissibility is still unclear. A positive RT-PCR result does not necessarily indicate the potential for viral transmission as this mode of testing can not discriminate between viable and inactive viral particles and furthermore readouts from different tests/commercial kits can not be easily compared[23–25].

In this context, there is a critical need to develop novel, more effective diagnostic tools capable of detecting active SARS-CoV-2 replication in the upper respiratory tract, which could provide useful indications for a more precise stratification of the patients and for the study of their potential infectivity. Current SARS-CoV-2 nucleic acid amplification diagnostic assays are qualitative or semi-quantitative, and since they can not discriminate genomic from subgenomic RNAs, do not provide any information concerning ongoing viral replication. In the light of all these considerations, subgenomic mRNAs could represent a relevant epidemiological target for a more specific diagnosis of COVID-19. Since these transcripts code for structural proteins required for assembly of novel viral particles, specific assays based on their detection may facilitate the development of novel diagnostic methods to better track the progression and to define robust guidelines for SARS-CoV-2 positive patients release.

In the current study, we present a method to accurately detect and quantify SARS-CoV-2 RNAs by ddPCR and meta-transcriptomics approaches. By using RNA extracted from nasopharyngeal swabs of SARS-CoV-2 positive subjects, we were able to specifically quantify the viral genome, by targeting the ORF1ab gene, and the N and S sgmRNAs.

Compared to qPCR, ddPCR allows a more precise absolute quantification of a target molecule without the need of a standard curve and, moreover, it shows a wide dynamic range with high sensitivity, even in the detection of low copy number of the target[26–29]. Several applications of ddPCR have been recently reported for the quantification of SARS-CoV-2 RNA genome, showing higher sensitivity, specificity, and reproducibility with respect to RT-qPCR[30–33].

In our study, we demonstrated that a significant reduction of subgenomic RNAs species is consistently detectable in samples with a low content of gRNA. Indeed, N and S smgRNAs were no longer quantifiable or barely detectable when gRNA copies/ng RNA were respectively less than 10 and 100. Samples with low viral gRNA content are thus characterized by very scarce or absent subgenomic transcripts, suggesting no or very limited SARS-CoV-2 transcription. Furthermore, regardless of the viral genome content of each sample, we demonstrated that N and S sgmRNAs have a specific expression pattern if compared to full-length gRNA, which remains almost constant in all analyzed samples and, consistent with the discontinuous mechanism of transcription of coronaviruses, is correlated to the proximity of the gene to the 3′ terminal portion of the genome.

Bioinformatics analyses of a subset of RNA samples, which were subjected to meta-transcriptomics sequencing, confirmed and extended results obtained by ddPCR on sgmRNAs expression. Indeed, the observed expression patterns for N and S sgmRNAs were largely consistent across all the samples, demonstrating that the approach used was highly reproducible and reliable. Moreover, when expression patterns of all sgmRNAs were considered, results were largely concordant with those reported by Kim et al.[18], suggesting that meta-transcriptomics sequencing can provide an accurate overview of the transcriptional dynamics of the genome of SARS-CoV-2. We observed a highly significant correlation between SARS-CoV-2 meta-transcriptomics sequencing data and ddPCR results, thus suggesting that approaches based on regression models could be effectively used to derive a plausible estimate of viral content for samples subjected to meta-transcriptomics sequencing. Even if only 2 out of 9 canonical sgmRNAs were quantified by ddPCR, the remarkable levels of correlation between sgmRNAs expression patterns as recovered by ddPCR and NGS data, suggest that expression profiles determined by our meta-transcriptomics approach are highly reliable and can provide useful indications for the study of SARS-CoV-2 replication/transcription in vivo.

Recently, Alexandersen et al. suggested that the presence of SARS-CoV-2 sgmRNAs in a sample, per se can not be considered an indicator of active replication, since these molecules can be protected from nuclease activity by double membranes structures, remaining detectable many days after infection[34]. Therefore, though sgmRNAs could be used as markers of ongoing active viral replication in cultured cells[18], it remains to be determined whether in SARS-CoV-2 positive patients sgmRNA levels correlate with disease severity, infectivity/transmission, and progression of the disease. In this contest, our ddPCR assays provide an approach to better address these questions, since more specific methods based on the relative quantification of sgmRNAs (with respect to the gDNA) could still, in principle, be used to follow COVID-19 progression and for the stratification of the patients. Anyhow, the results of our study seem to extend Alexandersen et al. conclusions, as we were able to detect sgmRNAs in patients with a low viral load, likely not infectious, and we demonstrated that the proportion of sgmRNAs to gRNA is decreased (Fig. 2) in these samples. In the light of these considerations, we believe that, if accurate quantitative methods are used for their quantification, in principle, sgmRNAs could be used to obtain information regarding the residual "of viral infectivity" and/or progression of the disease. For example, carefully designed assays based on the comparison of the levels of SARS-CoV-2 genomic and sub genomic RNAs could help clinicians both in monitoring COVID-19 progression and identifying the most appropriate therapeutic approach. Similar assays could also be used to gain a better understanding of the mechanisms of viral replication and propagation.

In conclusion, our results might contribute to the development of new effective strategies for SARS-CoV-2 RNAs detection,

supporting the currently available SARS CoV-2 diagnostic tools, for a more accurate stratification of patients, in particular of asymptomatic carriers, with relevant implications for the containment of COVID-19.

## Methods

**Viral RNA extraction.** Remnants from nasopharyngeal swabs (in UTM matrix) of 166 SARS-CoV2 positive subjects and 14 SARS-CoV-2 negative subjects were collected at the diagnostic laboratory of Ospedale Di Venere in Bari, from May to December 2020 (Table 1 and Supplementary Data 1). Ethical approval was not required for this study, as nasopharyngeal swab remnants were from subjects that remained unidentified and data generated concern exclusively viral sequences. 560 µL of UTM matrix were used to purify viral RNA using the QIAamp Viral RNA Mini Kit (Qiagen, Hilden, German) according to the manufacturer's instructions, without the addition of poly-A RNA carrier. The eluted RNA was treated with DNase (Zymo Research Corporation, Irvine, CA, USA) and successively concentrated with RNA Clean & Concentrator Kits (Zymo Research Corporation, Irvine, CA, USA), to the final volume of 15 µL, according to the manufacturer's instructions. RNA samples were quantitatively and qualitatively evaluated by NanoDrop 1000 (Thermo Fisher Scientific, Waltham, MA, USA) and 2100 Bioanalyzer (Agilent Technologies, Santa Clara, CA, USA) using the RNA 6000 Pico kit (Agilent Technologies), respectively. RNA Integrity Number (RIN) and DV200 value (the percentage of fragments >200 nucleotides), were calculated by Agilent Instrument Software. RNAs were stored at −80 °C until use.

**Synthetic SARS-CoV-2 genomic and sub-genomic DNA sequences.** DNA sequences corresponding to portions of ORF1ab, N and S genes, and N and S sgmRNAs were custom synthesized by GeneArt (Thermo Fisher Scientific, Waltham, MA, USA) (Supplementary Table 1) and employed as templates to evaluate accuracy, reproducibility, and sensitivity (LoD) of our SARS-CoV-2 ddPCR assays.

**RT-qPCR.** Before the DNAse treatment and the concentration step, RNA samples were analyzed by an in-house reverse transcription-qPCR assay using the primers/probe for E, N, and ORF1ab genes, as from the World Health Organization described by Corman et al.[35] to confirm the presence of the viral RNA. A 25 µL reaction was set up containing 5 µL of RNA, 12.5 µL of 2× reaction buffer provided with the Superscript III One Step RT-PCR system with Platinum Taq Polymerase (Invitrogen, Carlsbad, CA, USA), 1 µL of Reverse Transcriptase/Taq mixture, 10 µM of Forward and Reverse primers and 10 µM probe. Each assay was performed in triplicate on Applied Biosystems™ 7500 Real-Time PCR Systems (Thermo Fisher Scientific).

**SARS-CoV-2 RNAs quantification by ddPCR.** A forward primer in the leader sequence (LS) and a reverse primer in the S and N coding region, respectively, were designed to specifically detect SARS-CoV-2 S and N sgmRNAs; a primers pair was designed in the ORF1ab gene to detect the viral genome. Primer sequences are reported in Table 2. Each primer pair was initially evaluated by end-point PCR to confirm the size of the amplicon and to verify the presence of non-specific products by gel electrophoresis and, subsequently in the set-up phase of ddPCR, as recommended by the manufacturers, to define the optimal primer concentration and the most efficient annealing temperature. Amplicon's identity was confirmed by Sanger sequencing of the PCR product (Supplementary Fig. 2).

Variable amount (1.5–200 ng) of RNA was reverse transcribed in cDNA using the iScript™ Advanced cDNA Synthesis Kit for RT-qPCR (Bio-Rad, Hercules, CA, USA) and cDNA, diluted or as it is, was used as input in ddPCR experiments (up to 5 µl of cDNA per 22 µl reaction). The same cDNA preparation was used to perform the three ddPCR assays, for all samples. A reaction volume of 22 µl was prepared by combining cDNA (1–5 µl) with 11 µl of 2× Evagreen Supermix (Bio-Rad, Hercules, CA, USA), 220 nM ORF1ab primers or 250 nM N or S sgmRNA primers and water. The emulsion was produced in the QX200 Droplet Generator (Bio-Rad, Hercules, CA, USA) according to the manufacturer's instructions. Then the droplet-partitioned samples were amplified under the following thermal cycling conditions: for ORF1ab RNA: 1 cycle at 95 °C for 5 min, 40 cycles at 95 °C for 30 s and 60 °C for 1 min, 1 cycle at 4 °C for 5 min, 1 cycle at 90 °C for 5 min, final hold at 4 °C; for N sgmRNA: 1 cycle at 95 °C for 5 min, 40 cycles at 95 °C for 30 s and 58 °C for 1 min, 1 cycle at 4 °C for 5 min, 1 cycle at 90 °C for 5 min, final hold at 4 °C; for S sgmRNA: 1 cycle at 95 °C for 5 min, 40 cycles at 95 °C for 30 s and 56 °C for 1 min, 1 cycle at 4 °C for 5 min, 1 cycle at 90 °C for 5 min, final hold at 4 °C. Each RNA sample was analyzed at least in duplicate. For each experiment, a negative control (no template control) was used. Absolute quantification was performed using QuantaSoft version 7.4.1 software (Bio-Rad, Hercules, CA, USA) and the negative/positive thresholds were set manually, excluding samples with a number of droplets <10,000. QuantaSoft output results were expressed in copies/µl.

As variable RNA amounts and cDNA volumes were used, respectively, for RT and ddPCR experiments, absolute quantification of each target was obtained by calculating, firstly, the copies/µl of cDNA according to the Eq. (1):

$$copies/\mu l\ cDNA = \frac{dilution\ factor \cdot reaction\ volume\,(22\,\mu l) \cdot copies/\mu l}{\mu l\ cDNA\ in\ reaction} \quad (1)$$

The obtained number of copies/µl of cDNA was normalized for ng of RNA used as input in 20 µl of RT reaction volume (copies/ng RNA) (Supplementary Data 7).

By applying a hierarchical agglomerative clustering algorithm to ddPCR absolute quantification of SARS-CoV-2 genomic and sub-genomic RNAs, all samples were divided into three groups: the "high", the "middle" and the "low" group, composed of 21, 82, and 63 RNA samples, respectively. The estimated number of target copies were scaled by applying base 2 logarithm. Euclidean distances were computed by means of the dist() function as implemented by the R stat package[36]. Hierarchical clustering was performed by applying the hclust() function from the same software package. Finally, the cutree function was used to delineate three distinct clusters.

**Specificity and reproducibility of ORF1ab RNA and S and N sgmRNAs ddPCR assays.** The false-positive rate (FPR) was calculated on the results obtained from ddPCR assays performed on 20 no template controls (NTC) and 14 SARS-CoV-2 negative samples (Supplementary Data 2).

For samples in which positive droplets were detected, the FPR (%) was calculated as ratio FP/FP + TN, where: FP = false-positive events; TN = true negative events, in each well of the sample, as reported in other ddPCR studies[37–39]. The final FPR (%) was calculated as the average among all no-zero FPR values.

Reproducibility was evaluated on DNA sequences corresponding to portions of ORF1ab RNA, and N and S sgmRNAs by calculating the coefficient of variation inter- and intra-assay. For each ddPCR assay, at least eight replicates were analyzed in the same ddPCR to evaluate the intra-assay variability; the inter-assay variability was evaluated by performing three independent runs for every sample and target (Supplementary Data 3).

In order to evaluate the intra and inter-coefficient of variation (CV) we initially compured the CV per sample as follows:

$$CV_i = \frac{Sd_i}{m_i} \quad (2)$$

where $Sd_i$ and $m_i$ are the Standard deviation and the mean of the observed values of the sample i.

The inference of the inter-assay coefficient of variation was obtained as follow:

$$CV_{inter_i} = \frac{\sqrt{ns^2}}{M_i} \quad (3)$$

where, $n$ is the number of aliquots in each assay, $s^2$ is the variance of the mean of each day's result and $M_i$ is the overall mean for the sample I over all the performed assays.

**Accuracy and limit of detection of SARS-CoV-2 genomic and subgenomic ddPCR assays.** Ten serial dilutions of custom synthetic DNA sequences corresponding to ORF1ab gRNA and N and S sgmRNAs were employed to evaluate accuracy and LoD of each ddPCR assay. Each DNA template was diluted to 1 ng/µl and quantitatively analyzed by NanoDrop 3300 fluorospectrometer (Thermo Fisher Scientific) using the Quant-iT PicoGreen dsDNA Assay Kit (Invitrogen). The expected copy number contained in 1 ng of template was calculated considering the sequence length (250 bp), the molecular weight of the molecule, and finally, the moles number corresponding to 1 ng. For each dilution point, at least nine replicates were analyzed. The accuracy was evaluated plotting expected *vs* observed target copies per reaction for each dilution point; coefficient of determination ($R^2$) of SARS-CoV-2 target quantification was assessed by linear regression analysis (MedCalc statistical software, version 19.6.4). The Limit of Detection (LoD) of ORF1ab, sgN, and sgS ddPCR assays was defined by probit analysis on the same RNA dilutions used for accuracy assessment. All data are reported as copies/reaction (Supplementary Data 4–6).

**Meta-transcriptomics sequencing of SARS-CoV-2 genome and data analysis.** 113 RNA samples were used to sequence SARS-CoV-2 genome, using the meta-transcriptomics approach. Libraries were prepared using the Truseq Stranded Total RNA with Ribo Zero plus protocol (Illumina, San Diego, CA, USA) with some changes correlated to the quantity and quality of RNA extracted. Firstly, according to the availability of each RNA, a variable amount of total RNA (5–100 ng) was used as input for the library preparation. As the quality of RNA was not high (RIN ranging from 2 and 5, DV200 value > 30%), the incubation time of the RNA fragmentation step at 94 °C was decreased to 25 s. Finally, the number of cycles in the final enrichment PCR step varied from 15 to 18, depending on the quantity of total RNA used as input, as suggested by the Illumina Truseq Guide: for RNA input range 40–100 ng, the number of cycles was set to 15, for input range 20–40 ng, the number of cycles was set to 17, for input <20 ng, the number of cycles was set to 18. Libraries were sequenced on NextSeq500 platform (Illumina, San Diego, CA, USA) to generate 2 × 75 bp paired-end (PE) reads. 1% of the PhiX genome library was loaded in each run. An average of 13 M PE reads were generated per sample (Supplementary Data 8). SARS-CoV-2 genome assemblies were performed by means of the "Assembly of SARS-CoV-2 from pre-processed reads" workflow as available from the COVID-19 Galaxy[40].

**Detection and quantification of sgmRNA junction reads.** Annotation of SARS-CoV-2 sgmRNA was obtained from http://hgdownload.soe.ucsc.edu/

downloads.html in the form of a gtf file. For every subgenomic transcript, the corresponding subgenomic junction sequence was reconstructed by *in silico* juxtaposition of the LS in the 5′ UTR with the first 70 residues of each gene. For the S and N sgmRNAs, the sequence of the amplicon targeted by the ddPCR assay was used. Similarly, a region corresponding to the ORF1ab amplicon, was used to count metatranscriptomics reads associated with ORF1ab and to quantify gRNA. Meta-transcriptomics reads were mapped to the ORF1ab target sequence and to the complete collection of subgenomic junction sequences with the Bowtie2[41] software, using the —sensitive preset. A custom Perl script was used to count the number of reads associated with each target region and to obtain a table of counts. Counts were log scaled, with base 2 logarithm. Graphical representation of the data was performed by using the boxplot function as available in the standard library of the R programming language. Correlation analyses were performed by means of the trendline function from the basicTrendline R package. Only samples with more than 100 reads mapping on the SARS-CoV-2 viral genome (110 out of 113 sequenced samples) were considered in the analysis.

**Statistics and reproducibility**. DdPCR assays accuracy was analyzed by linear correlation comparing expected vs obtained copies for each SARS-CoV-2 RNA target in R. LoD (SARS-CoV-2 copy number at a 95% detection rate) was calculated by probit analysis using MedCalc statistical software (version 19.6.4) on at least nine replicates in ten serial dilutions of custom synthetic DNA sequence for each analyzed target.

Kruskal–Wallis test was performed by GraphPad Prism 8.0.2 software (GraphPad Software, San Diego, CA, USA) for ddPCR SARS-CoV-2 targets analysis between different viral RNA content groups; for each RNA viral content group, we calculated N and S sgmRNAs absolute quantification as median of all RNA samples belonging to the same group with relative interquartile range (IQR).

A Mann–Whitney U test/Wilcoxon rank-sum test was performed to analyze N and S sgmRNA/ORF1ab expression ratio in the "high" viral RNA content group compared to "middle" and "low" viral RNA content groups. $p$-values < 0.05 were considered as statistically significant.

## Data availability

The authors declare that the main data supporting the findings of this study are available within the article and its Supplementary Information file. Source data underlying figures are provided in Supplementary Data. Genomic assemblies were deposited at GISAID. A complete list of GISAID accessions is provided in Supplementary Data 8.

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

## Acknowledgements

This work was supported by INTERREG V-A Regione Puglia "Study of viral clusters and gene variants related to the SARS-Cov-2 host response in the Apulia and Epirus region", MUR Project CMPT209085, and ELIXIR Italy. We thank Dr Rosanna Piluscio (UOC Patologia Clinica- PO "Di Venere" - ASL Bari) and Maria Rosa Mirizzi (IBIOM-CNR) for technical assistance.

## Author contributions

A.O., C.M., M.C., G.P. and A.M.D. conceived and designed the study; A.P., A.B., M.I. and M.d.A. performed RNA extractions and RT-qPCR experiments; A.O. performed ddPCR experiments; C.M. and E.N. performed meta-transcriptomics sequencing; M.C. performed sequencing data analysis; B.F. performed statistical analyses; A.M.D. and G.P. provided supervision. A.O., M.C., A.M.D. and G.P. wrote the manuscript. All authors reviewed and edited the manuscript.

## Competing interests

The authors declare no competing interests.
