## [Peer Review File · Communications Biology]

Reviewers' comments:

Reviewer #1 (Remarks to the Author):

The manuscript describes a ddPCR-based method to detect and quantify SARS-CoV-2 genomic ORF1ab and sgmRNAs for nucleocapsid (N) and spike (S) proteins. In support of the developed method, a good correlation was observed between subgenomic RNA expression levels obtained in ddPCR and those obtained by NGS metatranscriptomic sequencing.

The biological/clinical/diagnostic rationale behind the development of this assay is interesting, although the quantification of SARS-CoV-2 SgmRNAs described here does not directly prove the correlation between them and virus infectivity, as the study does not exploit an in vitro system. The manuscript is well written and the topic is clinically relevant. However, some points, especially regarding the assay developed and discussion, need to be addressed.

Below my major comments:

1) Accuracy and limit of detection (LoD) of ddPCR assays targeting SARS-CoV-2 genomic and subgenomic mRNAs.

In the manuscript, the accuracy of the fine-tuned assay is emphasized, but there is no data on the specificity of the assay. The assay should be tested in SARS-CoV-2 negative samples and shown to be selective against the presence of other viral sequences.

2) The performance of the developed assay is evaluated focusing on the accuracy of the method. It would be interesting to evaluate the performance of the assay by taking into account other parameters such as the reproducibility and the estimation of the intra- and inter-run variation coefficient.

3) A recent article (Alexandersen S et al., Nat Commun. 2020) suggests that the detection of subgenomic RNAs might not be a suitable indicator of active virus replication/transcription or active/recurrent infection as such RNAs persist long after active infection. Based on this new evidence, it would be interesting and necessary to discuss the results obtained in the manuscript in light of the above-mentioned paper. DOI: 10.1038/s41467-020-19883-7

4) Absolute quantification of SARS-CoV-2 ORF1ab mRNA and N and S sgmRNAs by ddPCR. In this session, the median and IQR values of the quantifications of the 3 genes should be reported.

Please add the full IQR range. Normally the 25^o percentile and the 75^o percentile are reported for the IQR, why is only one value reported here? And does the reported value correspond to the 75^o percentile?

5) Lines 138-142: "As shown in Figure 2c-d, N and S sgmRNAs had a significant higher expression ratio in the "high" group if compared to the "middle+low" group (p-value = 0.0089 and 0.0002 respectively for N sgmRNA and S sgmRNA)..."

Why was it chosen to combine the middle and low groups for this analysis? And not were the individual groups compared? (high vs. middle, high vs. low and middle vs. low). What is the criterion for combining the middle and low groups?

The set up analysis might be unbalanced because the groups stratified as high vs middle+low are not uniform and there is a difference in sample size: 22 high vs 145 middle+low.

6) Lines 143-144: Finally, based on the assumption that ~ 20 picograms of RNA are the average content of a mammalian cell¹⁷, the expected number of N and S sgmRNAs per cell were calculated.

For the sake of clarity of interpretation, it would be appropriate to briefly explain the reason for the conversion of values into copies/cells.

7) Lines:178-181 "Furthermore, by NGS analysis and consistently with our ddPCR data of N and S sgmRNAs relative expression, a significantly higher subgenomic to genomic RNA ratio was observed for all canonical sgmRNAs, in the "high" RNA content group if compared to "middle"+"low" groups (Supplementary Figure 2).

See comment 4 regarding the lines 138-142.

8) SARS-CoV-2 RNAs quantification by ddPCR

For ddPCR RNA quantification, a dual step was chosen, first RT and then PCR, why? Why was a one-step not performed?

9) Moreover, for quantification, ddPCR allows the use of both 2x EvagreenSupermix and Supermix for probes. Why was the former chosen? Is the yield of the experiments better?

Minor Comments:

10) Table 1. RNA samples from COVID-19 confirmed anonymized subjects employed in the study. It is suggested to change the table's title as it is not completely clear and appropriate to the content of the table.

Also, please clarify how the age is reported; mean \pm SD?

11) Table 2. N and S sgRNA molecules number per cell according to viral RNA content group. Please clarify how the values have been reported in the table. Mean or median values?

12) Figure 2 b. Please insert in the "low" group the black lines representing the median and IQR as report in the other graphs.

13) Figure 2 Legend.

The number of samples in the "low" group should be 63 and not 62. Please correct this point.

14) COVID-19 positive subjects

It is more appropriate to define to them as SARS-CoV-2 positive subjects.

15) Detection and quantification of sgRNA junction reads and Meta-transcriptomics results are expressed on a logarithmic scale with base 2 logarithm. Why not base 10 logarithm?

16) In supplementary table 1 the ddPCR primers are reported. For transparency, the authors should complete this information with the sequences of the probes used.

Reviewer #2 (Remarks to the Author):

Communications Biology: "Accurate detection and quantification of SARS-CoV-2 genomic and subgenomic mRNAs by ddPCR and meta-transcriptomics analysis"

In this study titled "Accurate detection and quantification of SARS-CoV-2 genomic and subgenomic mRNAs by ddPCR and meta-transcriptomics analysis," the authors describe the development of ddPCR-based assays designed to measure one genomic and two subgenomic mRNA targets and their application to a large number of clinical samples. This work focuses on an important topic of SARS-CoV-2 research. While the use of ddPCR assays or next generation sequencing to detect SARS-CoV-2 subgenomic RNAs is not novel, the application of these assays to a large clinical cohort is of interest to researchers studying SARS-CoV-2 transcription. Strengths of the paper include the application of complementary techniques (ddPCR and NGS) to measure subgenomic transcripts and the large number of clinical samples. Weaknesses include the lack of clinical information from the samples and significant concerns regarding the validation of the ddPCR assays, particularly with regard to the circular manner in which the sensitivity (LOD) was measured and the lack of data to show the specificity of the assays for subgenomic transcripts or even for SARS-CoV-2. These and other concerns regarding the study design, technical rigour, and inclusion of additional data should be addressed prior to publication.

Major comments:

Lines 60-62: Some clinical assays target genes in ORF1ab, and thus may be specific for genomic RNA.

63-66, 74-77, 150-153, 245-250: The authors discuss benefits of tools that can detect active SARS-CoV-2 replication, however the data presented in this study do not directly address this. It is controversial as to whether sgRNAs are markers of active replication (Alexanderson et al Nature Communic 2020). As this study focuses on the development and measurement of subgenomic RNAs, it is worthwhile addressing counter arguments in the field that suggest that subgenomic RNAs are not markers of active replication and how this relates to the assays described here.

100-102: There are no details regarding how the forward and reverse primers were designed. Though it may be coincidental given the short leader sequence of SARS-CoV-2, the forward primer for N sgRNA, "LS_for1" is largely the same as (only 2 nucleotides shorter than) the forward primer for subgenomic E reported in Wölfel et al., Nature 2020. Furthermore, the table listing these primers "Supplementary Table 1", should be in the main manuscript for accessibility. This table should also specify that the primers are listed in 5'-3' orientation and should note the SARS-CoV-2 coordinates.

102-103 and 301-302. No data is presented to show the specificity or false positive rates for the ddPCR assays. What was used as the negative control(s)? ddPCR is notorious for having a high false positive rate; the published rate is about 1/3. Moreover, it appears that the authors' ddPCR assays use Evagreen instead of a probe, which may reduce cost but may also reduce specificity. What are the false positive rates on human cellular RNA? How about the false positive rates for the subgenomic assays when applied to genomic SARS-CoV-2 RNA? Do the subgenomic assays detect genomic RNA, especially when the latter is present at much higher copies? This data should be shown.

105-108, 109-115, 321-329: How were the expected copy numbers determined for the subgenomic RNAs? The data supporting the accuracy and LOD of the SARS-CoV-2 subgenomic and genomic RNA assays seem to be based entirely on dilution of one clinical sample with a high viral load (Ct= 14). It is unclear how the clinical sample was externally validated/quantified: methods only mention "by RT-qPCR". It is conceivable that expected copy numbers for genomic RNA could be estimated from Ct values from RT-qPCR. However, it is unclear how the expected copies of subgenomic RNAs were calculated from a clinical sample. It seems that the "expected" values were actually measured by applying the authors' new ddPCR assays to the undiluted clinical sample, which they then diluted to determine the measured values. If so, the measured copies and LOD would just be a function of how accurately they diluted the sample. This method is circular and cannot be used to determine the true sensitivity or LOD. For example, suppose the clinical sample has 10,000 copies of subgenomic S but the new ddPCR assay only detects 1,000 copies. If you then dilute this sample by 1,000-fold and detect one copy, you might think your assay has a detection limit of 1 copy, but you are actually detecting 10 copies. In order to truly measure the sensitivity and LOD, you would need standards for the genomic and two subgenomic targets, and you would need to have an independent means to quantify the copy numbers in the standards. More rigorous assessment of the assay performance is required to conclude that the assays are both sensitive and accurate for quantification.

Generally, considerably more validation data should be included. Examples include 1D or 2D ddPCR plots. These can give an indication as to whether there is good separation between positive and negative droplets, from which it is possible to infer that primer concentrations and experimental conditions are appropriate.

123-130. If these three groups were stratified based on levels of genomic and subgenomic RNA, it seems expected (and not particularly interesting) that levels of subgenomic N and S would differ between the groups. On the other hand, it is not clear that you would expect differences between groups in the ratio of subgenomic to genomic RNA, which is interesting. Aside from stratification of samples into 'high', 'medium', and 'low' RNA content groups, there is no indication of how this correlates to the course of infection in individuals or with active viral replication. What is the clinical significance of the 3 groups expressing different levels of sgRNA? Is it possible to report when the samples were collected? Whether sample collection occurred close to symptom onset or later in the course of infection would likely contribute to differences observed in SARS-CoV-2 RNA content. This may be a useful correlation if these data are available.

336-339: This sentence is not clear. Was the RNA quality low for all samples? This can affect the quality of sequencing results and can lead to biases.

342-343: How were quality and quantity determined?

Minor comments:

157-159: Next generation sequencing is well known to have biases, with varying coverage (by read numbers) of different regions that are expected to be in the same transcripts.

167-169: Why is the correlation much lower for subgenomic S? This could be due to bias in the read coverage for sequencing or differences between the sensitivity of the assays for subgenomic S versus subgenomic N or genomic ORF1ab.

173: Authors note that they observe sgmRNAs encoded by the 3' end of the genome displayed higher levels of expression in line with "other studies", but only reference one study.

172-176: Other studies have shown a different pattern for the abundances of subgenomic RNAs (Kim et al Cell 2020, Nomburg et al Genome Med 2020, Alexanderson et al Nature Commun 2020, Doddapaneni BioRxiv 2020, Telwatte et al Methods 2021).

Why is there a higher proportion of sgmN relative to gRNA in the 'low' RNA content group compared to the 'middle' group (Fig. 2c)? This also holds true for some sgmRNAs from the metatranscriptomic data (Supplementary. Fig 2) but not all.

263-265: The authors note that poly-A RNA carrier was not added during RNA extraction- does this assist in viral RNA recovery?

265-267: To what final volume was the original 560uL VTM concentrated following extraction and use of Zymo Concentrator kit (i.e. what fraction does the 5uL RNA input into RT reactions represent)?

290-291: What does this sentence mean?

294-300: Variable annealing temperatures are used for ORF1ab, sgmN and sgmS. There are no data or description in the methods to indicate why these conditions were selected. Does the lower annealing temperature for sgmS still avoid any non-specific amplification?

340: Setted?

Figure 1a-c: What are the dilutions to the right? The colour scheme makes it difficult to distinguish dilutions used. Also, the axes should be reversed, so that observed is plotted as a function of expected and the y in the formula shows how observed relates to expected.

Fig 2: If there are statistically significant differences between groups, these should be indicated in the figure.

Fig 4: Clarify if "genes" on the y axis refer to total reads from each coding region or to specific subgenomic reads containing each coding region.

Table 2: Reporting levels of ORF1ab in these groups would be useful to gauge levels of genomic RNA.

Supplementary Table 1: there is a typo in heading "amplicon length".

Reviewer #3 (Remarks to the Author):

The authors have designed elegant and robust methods to analyze the expression of SARS-CoV2 at the mRNA level. They show that there is a more or less consistent ratio between the expression of N and S as compared to the number of genome copies. Two methods are used that complement and corroborate each other.

A point of concern is that it is not well described how the samples are obtained. "Remnants of swabs were collected..." In these samples a lot can happen to mRNA in cells, whereas the genomic RNA in virus particles remains relatively stable. The conditions and duration of storage of the samples must be described more accurately to assess the possible impact on the mRNA. The problem to keep mRNA intact may have caused a severe underestimation of mRNA levels. As the authors show in the last figure where all mRNAs are evaluated, that the mRNAs fall below the level of the genomic RNA. This contradicts earlier findings of mRNA levels in cultured cells, where mRNA levels are much higher than genomic RNA levels.

It is difficult to compare cells in tissue culture with cells in swabs but there should be some correlation.

In my own experience I have seen strong competition of mRNA and overlapping genomic regions, both in metagenomic reads and in PCR reactions.

Another personal observation is that there can be big differences between sample in the ratio of other mRNAs than the ones studied here. For example mRNAs 6 and 7 have very high expression levels in some samples.

The authors should indicate very clearly that one of the limitations of this work is the fact that they restricted their complete analysis to only 2 of the 8 or 9 mRNAs available, and that there are still several aspects to be investigated.

Manuscript COMMSBIO-21-0911

Reply to Reviewer 1

We thank Reviewer 1 for his/her remarkable observations.
Below the answers to all comments.

Reviewer #1 (Remarks to the Author):

The manuscript describes a ddPCR-based method to detect and quantify SARS-CoV-2 genomic ORF1ab and sgmRNAs for nucleocapsid (N) and spike (S) proteins. In support of the developed method, a good correlation was observed between subgenomic RNA expression levels obtained in ddPCR and those obtained by NGS metatranscriptomic sequencing. The biological/clinical/diagnostic rationale behind the development of this assay is interesting, although the quantification of SARS-Cov-2 SgmRNAs described here does not directly prove the correlation between them and virus infectivity, as the study does not exploit an in vitro system. The manuscript is well written and the topic is clinically relevant. However, some points, especially regarding the assay developed and discussion, need to be addressed. Below my major comments:

1) Accuracy and limit of detection (LoD) of ddPCR assays targeting SARS-CoV-2 genomic and subgenomic mRNAs.

In the manuscript, the accuracy of the fine-tuned assay is emphasized, but there is no data on the specificity of the assay. The assay should be tested in SARS-CoV-2 negative samples and shown to be selective against the presence of other viral sequences.

As suggested, additional experiments were performed to evaluate the specificity of our ddPCR assays. False Positive Rates (FPR) were estimated by using 20 no template controls (NTC) samples and 14 SARS-CoV-2 negative RNA samples. As shown in the Supplementary Data 2, the ddPCR assays for the quantification of ORF1ab RNA S and N sgmRNAs had very low FPR values and hence are highly specific in the detection of viral sequences. Furthermore, we performed ddPCR assays using custom “off-target” DNA sequences of the SARS-CoV-2 N and S genes, as a template. Again, no amplification and thus no sgmRNAs were detected (Supplementary Fig.1). Finally, to further confirm the assay's specificity, we performed Sanger sequencing of ORF1ab, N and S sgmRNA PCR amplicons generated from two SARS-CoV-2 positive RNA samples. Sequences obtained correspond to each target (Supplementary Fig. 2).

2) The performance of the developed assay is evaluated focusing on the accuracy of the method. It would be interesting to evaluate the performance of the assay by taking into account other parameters such as the reproducibility and the estimation of the intra- and inter-run variation coefficient.

We estimated intra- and inter-assay CV % and we reported all data in results and methods sections (Lines 125-129 and Supplementary Data 3).

3) A recent article (Alexandersen S et al., Nat Commun. 2020) suggests that the detection of subgenomic RNAs might not be a suitable indicator of active virus replication/transcription or active/recurrent infection as such RNAs persist long after active infection. Based on this new evidence,

it would be interesting and necessary to discuss the results obtained in the manuscript in light of the above-mentioned paper. DOI: 10.1038/s41467-020-19883-7.

We agree with the referee that the results reported in our work seem to contradict at least partially the main findings of Alexandersen et al, however in our opinion the aim of the two studies are different and the conflict is only apparent. Indeed, the study performed by Alexandersen et al is mostly qualitative and the main message is that the presence of sgmRNA, in a sample per-se, is not an indicator of active viral replication. On the other hand, our approach is quantitative: we observe that notwithstanding the fact sgmRNAs can be detected even in samples with a high Ct, where the levels of genomic (gRNA) and subgenomic (sgmRNA) viral RNAs are compared, lower Ct values (and potentially active viral replication) are associated with a higher ratio of sgmRNA to gRNA (Figure 2). In conclusion, our results indicate higher load of sgmRNA in low Ct vs middle and high Ct samples. A brief comment concerning the interpretation of the results in the context of the main findings by Alexandersen et al is included in the revised version of the manuscript.

Additionally, the two studies differ in several methodological aspects, which cannot be overlooked:

- Alexandersen et al. quantified only the 7a sgmRNA in a limited number of RNA samples, from patients with minor clinical manifestations. In this study we analyzed more than 150 RNA samples from confirmed SARS-CoV-2 positive individuals, with variable viral RNA loads and we quantified 2 different (N and S) sgmRNAs.
- A different RNA extraction protocol was used: in Alexandersen et al, “nucleic acid extraction was performed by heating extracted nucleic acids at 70 °C for 5 min and rapid cooling on ice before cDNA synthesis”; in our work, RNA extraction was performed by QIAamp Viral RNA Mini Kit which uses highly denaturing conditions and does not require nuclease activity.
- NGS sequencing in Alexandersen et al paper was performed by amplicon sequencing, a technique that is not considered optimal for the quantification of sgmRNAs (as no specific primers are included in the design) and can introduce systematic biases; in our study, all the samples were sequenced using a meta-transcriptomic approach which does not capture specific target sequences and is considered to be more reliable for the quantification of the abundance of RNAs in a sample.

Finally, our results are consistent with those reported by Wölfel et al. mentioned in the text;

- A recent study that analyzed E sgmRNA in convalescent and antibody-treated rhesus macaques in order to discriminate input challenge and replicating virus for assessing the protective efficacy of natural or vaccine-mediated immunity. The study pointed out the importance of sgmRNA in highlighting active replicating viruses even though it was tested in a not-human model (doi:10.1128/JVI.02370-20).

In light all these considerations, we conclude that while there are no conclusive evidences to attest that detection of viral sgmRNAs in a sample demonstrates ongoing viral replication, we speculate that, in the light of our and other concordant published studies, carefully designed assays based on the comparison of the levels of SARS-CoV-2 genomic and sgmRNAs could help clinicians both in monitoring the progression of the infection, identify the most appropriate therapeutic approach and the efficacy of vaccines. Similar assays could also be used to better comprehend the viral mechanisms of replication and propagation.

4) Absolute quantification of SARS-CoV-2 ORF1ab mRNA and N and S sgmRNAs by ddPCR. In this session, the median and IQR values of the quantifications of the 3 genes should be reported.

Please add the full IQR range. Normally the 25° percentile and the 75° percentile are reported for the IQR, why is only one value reported here? And does the reported value correspond to the 75° percentile?

We corrected the IQR range: previously we reported IQR as the difference between 75° and 25° percentiles. In the revised version we fully reported all quartiles details and IQR (see line 158 and followings).

5) Lines 138-142: “As shown in Figure 2c-d, N and S sgmRNAs had a significant higher expression ratio in the “high” group if compared to the “middle+low” group (p-value = 0.0089 and 0.0002 respectively for N sgmRNA and S sgmRNA)...”

Why was it chosen to combine the middle and low groups for this analysis? And not were the individual groups compared? (high vs. middle, high vs. low and middle vs. low). What is the criterion for combining the middle and low groups?

The low and middle group were combined since no statistically significant differences were observed between the two. We have repeated all the analyses by performing all the tests between all the different groups. We confirm that the expression ratio of both S and N sgmRNAs are significantly higher in the “high” compared with any of the other 2 groups. P-values are indicated in the corresponding figures (see captions) and in the methods and results sections.

The set up analysis might be unbalanced because the groups stratified as high vs middle+low are not uniform and there is a difference in sample size: 22 high vs 145 middle+low.

To the best of our knowledge this should not be an issue for the statistical tests used in this study. The Wilcoxon sum and rank test and the Kolmogorov Smirnov test are not biased by the number of instances contained in the 2 groups to be compared and do not require groups of matched size.

6) Lines 143-144: Finally, based on the assumption that ~ 20 picograms of RNA are the average content of a mammalian cell¹⁷, the expected number of N and S sgmRNAs per cell were calculated. For the sake of clarity of interpretation, it would be appropriate to briefly explain the reason for the conversion of values into copies/cells.

We calculated the expected copy number of sgmRNAs per cell because, even though it is a speculation, we believed that this data could provide useful information about the intracellular viral content. We expressed this quantity as copies/cells and this value provides direct evidence on the amount of sgmRNAs present in a cell.

7) Lines:178-181 “Furthermore, by NGS analysis and consistently with our ddPCR data of N and S sgmRNAs relative expression, a significantly higher subgenomic to genomic RNA ratio was observed for all canonical sgmRNAs, in the “high” RNA content group if compared to “middle”+“low” groups (Supplementary Figure 2).

See comment 4 regarding the lines 138-142.

The analyses were repeated without aggregating the samples assigned to the low and middle groups. A new version of the Supplementary Figure 4 was presented which includes 2 p-values: 1) comparison of “high” with “low”; 2) comparison of “high” with “middle”.

8) SARS-CoV-2 RNAs quantification by ddPCR

For ddPCR RNA quantification, a dual step was chosen, first RT and then PCR, why? Why was a one-step not performed?

Due to the low concentrations of RNA samples extracted from the naso-pharyngeal swabs, we chose the dual step RT-PCR approach because it allowed the possibility to use, for each sample, the same cDNA for the three ddPCR assays.

9) Moreover, for quantification, ddPCR allows the use of both 2x EvaGreenSupermix and Supermix for probes. Why was the former chosen? Is the yield of the experiments better?

We chose the DNA binding dye (EvaGreen) assay because we evaluated that it provided a good target specificity and accuracy.

Minor Comments:

10) Table 1. RNA samples from COVID-19 confirmed anonymized subjects employed in the study. It is suggested to change the table's title as it is not completely clear and appropriate to the content of the table.

Also, please clarify how the age is reported; mean \pm SD?

Done.

11) Table 2. N and S sgRNA molecules number per cell according to the viral RNA content group. Please clarify how the values have been reported in the table. Mean or median values?

Done.

12) Figure 2 b. Please insert in the "low" group the black lines representing the median and IQR as reported in the other graphs.

Median of the “low” group has value 0. We performed graphs by GraphPad Prism and, when the Y axis is logarithmic or in scientific notation, it simply can't plot it, as specified in the Figure 2 caption.

13) Figure 2 Legend

The number of samples in the "low" group should be 63 and not 62. Please correct this point.

Done.

14) COVID-19 positive subjects

It is more appropriate to define to them as SARS-CoV-2 positive subjects.

Corrected.

15) Detection and quantification of sgRNA junction reads and Meta-transcriptomics results are expressed on a logarithmic scale with base 2 logarithm. Why not base 10 logarithm?

We chose base 2 logarithm because normally base 2 logarithms are used to scale/normalize gene expression data and or changes in their expression (i.e see log₂ fold changes for example). A doubling (or the reduction to 50%) is often considered as a biologically relevant change. On the log₂ scale this translates to one unit (+1 or -1). That's a simple value, easy to recall, and it is more "fine grained" than using higher bases (like log₁₀).

16) In supplementary table 1 the ddPCR primers are reported. For transparency, the authors should complete this information with the sequences of the probes used.

We did not use probes.

Reply to Reviewer 2

We thank Reviewer 2 for his/her remarkable observations.

Below the answers to all comments.

Reviewer #2 (Remarks to the Author):

Communications Biology: "Accurate detection and quantification of SARS-CoV-2 genomic and subgenomic mRNAs by ddPCR and meta-transcriptomics analysis"

In this study titled "Accurate detection and quantification of SARS-CoV-2 genomic and subgenomic mRNAs by ddPCR and meta-transcriptomics analysis," the authors describe the development of ddPCR-based assays designed to measure one genomic and two subgenomic mRNA targets and their application to a large number of clinical samples. This work focuses on an important topic of SARS-CoV-2 research. While the use of ddPCR assays or next generation sequencing to detect SARS-CoV-2 subgenomic RNAs is not novel, the application of these assays to a large clinical cohort is of interest to researchers studying SARS-CoV-2 transcription. Strengths of the paper include the application of complementary techniques (ddPCR and NGS) to measure subgenomic transcripts and the large number of clinical samples. Weaknesses include the lack of clinical information from the samples and significant concerns regarding the validation of the ddPCR assays, particularly with regard to the circular manner in which the sensitivity (LOD) was measured and the lack of data to show the specificity of the assays for subgenomic transcripts or even for SARS-CoV-2. These and other concerns regarding the study design, technical rigour, and inclusion of additional data should be addressed prior to publication.

Major comments:

Lines 60-62: Some clinical assays target genes in ORF1ab, and thus may be specific for genomic RNA.

We agree and we have amended the text accordingly.

63-66, 74-77, 150-153, 245-250: The authors discuss benefits of tools that can detect active SARS-CoV-2 replication, however the data presented in this study do not directly address this. It is controversial as to whether sgRNAs are markers of active replication (Alexanderson et al Nature Communic 2020). As this study focuses on the development and measurement of subgenomic RNAs, it is worthwhile addressing counter arguments in the field that suggest that subgenomic RNAs are not markers of active replication and how this relates to the assays described here.

This is a remarkable point we now address in the Discussion- See also answers to Reviewer #1 for a more detailed explanation.

100-102: There are no details regarding how the forward and reverse primers were designed. Though it may be coincidental given the short leader sequence of SARS-CoV-2, the forward primer for N sgmRNA, “LS_for1” is largely the same as (only 2 nucleotides shorter than) the forward primer for subgenomic E reported in Wölfel et al., Nature 2020. Furthermore, the table listing these primers “Supplementary Table 1”, should be in the main manuscript for accessibility. This table should also specify that the primers are listed in 5’-3’ orientation and should note the SARS-CoV-2 coordinates.

At first, to detect the S and N sgmRNAs, we designed 2 different forward primers (LS_for1 and LS_for2) inside the leader sequence and different reverse primers inside the gene-specific sequence. During the PCR set-up phase, we performed all the possible combinations of forward/reverse primers to select the primers pair with best performance (not-specific products, no amplicons with NTC samples, etc). This is the reason why N and S sgmRNAs were detected with a different forward primer. Regarding the similarity of “LS_for1” primer with the forward primer for subgenomic E reported in Wölfel et al., Nature 2020, this is coincidental, because of the short length of the leader sequence. As requested, we have introduced a new Table (Table 3) in the main manuscript with the sequences and the SARS-CoV-2 genome coordinates of the primers.

102-103 and 301-302. No data is presented to show the specificity or false positive rates for the ddPCR assays. What was used as the negative control(s)? ddPCR is notorious for having a high false positive rate; the published rate is about 1/3. Moreover, it appears that the authors’ ddPCR assays use Evagreen instead of a probe, which may reduce cost but may also reduce specificity. What are the false positive rates on human cellular RNA?

We have evaluated the specificity of the ddPCR assay for each SARS-CoV-2 target by calculating the false positive rate (FPR) on 20 no template controls (NTC) and 14 SARS-CoV-2 negative RNA samples. These results are described now in the manuscript in a new paragraph “Assessment of specificity and reproducibility of designed SARS-CoV-2 ddPCR assay for genomic and subgenomic mRNAs (Lines 99-131 and Supplementary data 2). Furthermore, we have also included ddPCR 1D amplitude plots, for each SARS-CoV-2 target, performed on NTC samples, on SARS-CoV-2 negative RNA samples and on not specific DNA templates (Supplementary Figure 1). Although the DNA binding dye (EvaGreen) assay is considered less specific than hydrolysis probe, obtained results consistently indicate an high specificity of our assays (Supplementary data 2, Supplementary Figure 1).

How about the false positive rates for the subgenomic assays when applied to genomic SARS-CoV-2 RNA? Do the subgenomic assays detect genomic RNA, especially when the latter is present at much higher copies? This data should be shown.

We have performed ddPCR for N and S sgmRNAs using as template custom synthesized DNA sequences of the 5' region of N and S gene, respectively. ddPCR was performed in triplicate and no positive events have been detected by droplets analysis, confirming that the designed ddPCR assays are specific in detecting SARS-CoV-2 sgmRNAs (Supplementary Figure 1).

105-108, 109-115, 321-329: How were the expected copy numbers determined for the subgenomic RNAs? The data supporting the accuracy and LoD of the SARS-CoV-2 subgenomic and genomic RNA assays seem to be based entirely on dilution of one clinical sample with a high viral load (Ct= 14). It is unclear how the clinical sample was externally validated/quantified: methods only mention "by RT-qPCR". It is conceivable that expected copy numbers for genomic RNA could be estimated from Ct values from RT-qPCR. However, it is unclear how the expected copies of subgenomic RNAs were calculated from a clinical sample. It seems that the "expected" values were actually measured by applying the authors' new ddPCR assays to the undiluted clinical sample, which they then diluted to determine the measured values. If so, the measured copies and LOD would just be a function of how accurately they diluted the sample. This method is circular and cannot be used to determine the true sensitivity or LOD. For example, suppose the clinical sample has 10,000 copies of subgenomic S but the new ddPCR assay only detects 1,000 copies. If you then dilute this sample by 1,000-fold and detect one copy, you might think your assay has a detection limit of 1 copy, but you are actually detecting 10 copies. In order to truly measure the sensitivity and LOD, you would need standards for the genomic and two subgenomic targets, and you would need to have an independent means to quantify the copy numbers in the standards. More rigorous assessment of the assay performance is required to conclude that the assays are both sensitive and accurate for quantification.

We agree with the reviewer's criticisms. We have revised our approach by determining the sensitivity of each ddPCR assay using the previously described custom synthetic DNA sequences. The updated LoD data are reported in the main manuscripts in results and methods sections (Lines 133-147, Figure 1 and Supplementary Data 4-6).

Generally, considerably more validation data should be included. Examples include 1D or 2D ddPCR plots. These can give an indication as to whether there is good separation between positive and negative droplets, from which it is possible to infer that primer concentrations and experimental conditions are appropriate.

1D ddPCR plots have been included in Supplementary Figure 1.

123-130. If these three groups were stratified based on levels of genomic and subgenomic RNA, it seems expected (and not particularly interesting) that levels of subgenomic N and S would differ between the groups. On the other hand, it is not clear that you would expect differences between groups in the ratio of subgenomic to genomic RNA, which is interesting. Aside from stratification of samples into 'high', 'medium', and 'low' RNA content groups, there is no indication of how this correlates to the course of infection in individuals or with active viral replication. What is the clinical significance

of the 3 groups expressing different levels of sgRNA? Is it possible to report when the samples were collected? Whether sample collection occurred close to symptom onset or later in the course of infection would likely contribute to differences observed in SARS-CoV-2 RNA content. This may be a useful correlation if these data are available.

Unfortunately, as discussed in the manuscript, we used naso-pharyngeal swabs from anonymized patients and clinical information was not available. We agree with the reviewer that it would be very interesting to integrate/correlate additional data concerning the date of collection of the samples and the onset of the symptoms, but this was not the primary aim of our study.

336-339: This sentence is not clear. Was the RNA quality low for all samples? This can affect the quality of sequencing results and can lead to biases.

We agree with the reviewer that low quality of RNA can lead to biases in sequencing results. Unfortunately, the quality of RNA extracted from naso-pharyngeal swabs is known not to be high (Chiara et al, 2020 doi: 10.1093/bib/bbaa297). As suggested by Illumina technical note for RNA sequencing, we have calculated the DV200 value that evaluates the percentage of fragments > 200 nucleotides of the RNA preparation. As it was greater than 30% for all RNAs, they were considered suitable for sequencing analysis.

342-343: How were quality and quantity determined?

We included the qualitative and quantitative approaches in the Methods section of Viral RNA Extraction. In detail, RNA samples were quantitatively evaluated by microvolume UV-Vis Spectrophotometer technology using NanoDrop 1000 and qualitatively evaluated using the Agilent Bioanalyzer 2100 and RNA 6000 Pico kit.

Minor comments:

157-159: Next generation sequencing is well known to have biases, with varying coverage (by read numbers) of different regions that are expected to be in the same transcripts.

We agree with this comment of the reviewer. The most used method for SARS-CoV-2 genome sequencing (considering the low quality of the RNA input) is the amplicon/capture-based protocol that enriches the viral genome but it could have different efficiencies and result in coverage biases (Chiara et al, 2020 doi: 10.1093/bib/bbaa297). On the contrary, we used the meta-transcriptomic library preparation approach, which is considered to offer an unbiased representation of the RNAs present in a sample (Chiara et al, 2020 doi: 10.1093/bib/bbaa297; Xiao et al., 2020 doi: 10.1186/s13073-020-00751-4).

167-169: Why is the correlation much lower for subgenomic S? This could be due to bias in the read coverage for sequencing or differences between the sensitivity of the assays for subgenomic S versus subgenomic N or genomic ORF1ab.

In ddPCR assay, probably the lower correlation for S sgmRNA with respect to N sgmRNA and ORF1ab RNA could be due to a lower S sgmRNA expression respect to the other sgmRNAs (Kim et al., 2020) which results in a major difficulty in detecting S sgmRNA. Similarly, this is the reason why S sgmRNA is associated with a reduced number of reads in our sequencing.

173: Authors note that they observe sgmRNAs encoded by the 3' end of the genome displayed higher levels of expression in line with "other studies", but only reference one study.

Corrected.

172-176: Other studies have shown a different pattern for the abundances of subgenomic RNAs (Kim et al Cell 2020, Nomburg et al Genome Med 2020, Alexanderson et al Nature Commun 2020, Doddapaneni BioRxiv 2020, Telwatte et al).

Patterns of expression in these studies have been assessed by means of different approaches and sequencing technologies, applied to different biological samples. In the light of these considerations, we are not surprised that the patterns of expression are not completely consistent. The differences between the experimental design of the study and techniques used is briefly summarized below. However, we have noticed that, broadly speaking, patterns recovered by the different studies (included those reported in our study) are consistent with the discontinuous mechanism of transcription of coronaviruses and the most "expressed" transcripts correspond with those at the 3' terminal end of the genome (as it should be expected).

Kim et al (Cell 2020): infected Vero cells (not "real" patients) sequencing was performed by single molecule sequencing (Pac Bio), this technology generates long reads, and allows the study of "non-canonical" transcripts.

#Nomburg et al (Genome Med 2020): aggregates different single molecule sequencing datasets, all obtained from infected vero cells. Sequencing is again performed by single molecule sequencing (Pac bio).

#Alexanderson et al (Nature Commun 2020): included 12 samples, NGS sequencing performed by amplified+short read. Reads spanning the junctions of subgenomic reads in these settings are a by-product since the protocol does not include specific probes/primers (but see also previous answers).

Doddapaneni et al (BioRxiv 2020): used a custom oligonucleotide capture system to enrich for viral reads. Probes for sgmRNAs are included. It is well known that capture-based systems can introduce biases in the capture (and especially in the presence of genetic variants). The study included 45 SARS-CoV-2 clinical samples, but only 17 resulted in the reconstruction of a complete genome sequence and were used for the study of gene expression. The number is very limited and subgenomic RNAs are not detected for the majority of the genes.

#Telwatte et al (Methods. 2021): study is based on "a nasopharyngeal sample from a laboratory-confirmed SARS-CoV-2 acutely-infected individual (5 days post-symptom onset and 4 days after positive clinical PCR test)", quantification is performed by ddPCR not NGS sequencing.

Why is there a higher proportion of sgmN relative to gRNA in the 'low' RNA content group compared to the 'middle' group (Fig. 2c)? This also holds true for some sgmRNAs from the metatranscriptomic data (Supplementary. Fig 2) but not all.

In the middle and low groups, the sgmRNA copies are very low, and especially for the “low” group the variability is very high (as it can be observed from the whiskers of the boxplot). Therefore, some fluctuations/apparent inconsistencies can be observed. Some copy numbers estimated for N sgmRNA or ORF1ab were close to 0 and the ratio between the two could be changed significantly just by a small change in one or the other. A brief sentence was added to the text (Lines 179-182).

263-265: The authors note that poly-A RNA carrier was not added during RNA extraction- does this assist in viral RNA recovery?

We chose to avoid the use of the poly-A carrier during viral RNA extraction for the following reasons: 1) poly-A RNA carrier can contribute to overestimation the amount of the extracted RNA; 2) poly-A RNA carrier might compete for the binding to oligo(dT) primers used, with random primers, in the reverse transcription step, before ddPCR assay and RNA-seq library preparation.

265-267: To what final volume was the original 560uL VTM concentrated following extraction and use of Zymo Concentrator kit (i.e. what fraction does the 5uL RNA input into RT reactions represent)? We have reported the final volume of the RNA preparation after the Zymo Concentrator kit (Line 324).

290-291: What does this sentence mean?

We have explained this point better in the manuscript (Lines 348-355).

Each primer pair was firstly evaluated by end-point PCR to confirm amplicon's size and to check for the presence of non-specific products. Then, we carried out a set-up phase of ddPCR, as recommended by the manufacturers, to define the optimal primer concentration and the more efficient annealing temperature for each assay.

294-300: Variable annealing temperatures are used for ORF1ab, sgmN and sgmS. There is no data or description in the methods to indicate why these conditions were selected. Does the lower annealing temperature for sgmS still avoid any non-specific amplification?

The ddPCR assay requires a set-up phase for obtaining high quality data. This is the reason why the annealing temperature is variable for the three targets.

340: Setted?

It was a typo in the heading: it has been corrected.

Figure 1a-c: What are the dilutions to the right? The colour scheme makes it difficult to distinguish dilutions used. Also, the axes should be reversed, so that observed is plotted as a function of expected and the y in the formula shows how observed relates to expected.

Done

Fig 2: If there are statistically significant differences between groups, these should be indicated in the figure.

Done.

Fig 4: Clarify if “genes” on the y axis refer to total reads from each coding region or to specific subgenomic reads containing each coding region.

“Genes” refer to subgenomic RNA reads. It has been clarified.

Table 2: Reporting levels of ORF1ab in these groups would be useful to gauge levels of genomic RNA.

We have included the copies of gRNA in the Table, now Table 3.

Supplementary Table 1: there is a typo in heading "amplicon length".

Corrected, this Table has been moved in the main manuscript (now Table 2).

REVIEWERS' COMMENTS:

Reviewer #1 (Remarks to the Author):

Authors properly addressed all the points raised. I don't have any further comments.

Reviewer #2 (Remarks to the Author):

In general, the authors were very responsive to the reviewers' questions. I have only two minor comments:

104-114: It is great that the authors list both the actual numbers of samples that had false positives (out of the total) and the numbers of false positive droplets. However, it is not clear how they calculated their false positive rates. For example, when 2 out of 20 samples had false positives, they list a rate of 0.0067%. How was this % obtained? Was it the % of all droplets that were false positives? The false positive rate is usually expressed as the fraction or % of samples that gave false positives, not the % of droplets. Also, it looks like the percentages of samples that gave false positives were actually quite high: 10-20% for no-template controls (presumably water) and up to 11/14 for SARS-CoV-2 negative RNA. It is nice that the absolute number of false positive droplets was low. However, to interpret the possible contribution of false positives to the results from the clinical samples, which are expressed as copies/ng RNA, it would be helpful to know the range in the number of positive droplets per well (best) or the range of RNA inputs per well.

414-418: It is somewhat unclear what was done here. Did the authors measure the DNA concentration again using the Quant-iT PicoGreen dsDNA Assay Kit? Did they then use the DNA concentration from this kit and the calculated molecular weight to determine the number of molecules per μL ? Was this sample diluted further to get down to 1 copy?